# Pharmacophore-Conditioned Diffusion Model for Ligand-Based *De Novo* Drug Design

## Abstract

Developing bioactive molecules remains a central, time- and cost-heavy challenge in drug discovery, particularly for novel targets lacking structural or functional data. Pharmacophore modeling presents an alternative for capturing the key features required for molecular bioactivity against a biological target. In this work, we present PharmaDiff, a pharmacophore-conditioned diffusion model for 3D molecular generation. PharmaDiff employs a transformer-based architecture to integrate an atom-based representation of the 3D pharmacophore into the generative process, enabling the precise generation of 3D molecular graphs that align with predefined pharmacophore hypotheses. Through comprehensive testing, PharmaDiff demonstrates superior performance in matching 3D pharmacophore constraints compared to ligand-based drug design methods. Additionally, it achieves higher docking scores across a range of proteins in structure-based drug design, without the need for target protein structures. By integrating pharmacophore modeling with 3D generative techniques, PharmaDiff offers a powerful and flexible framework for rational drug design.

## 1 Introduction

Computer-Aided Drug Discovery (CADD) aims to identify novel therapeutics against desired targets by investigating molecular properties using computational tools and available databases. A pharmacophore is a CADD method first introduced by Ehrlich (1909) and is defined by the IUPAC as "an ensemble of steric and electronic features necessary to ensure the optimal supramolecular interactions with a specific biological target and to trigger (or block) its biological response." (Wermuth et al., 1998). A 3D pharmacophore hypothesis also accounts for the arrangement of these chemical features relative to each other in 3D space (Zhu et al., 2023a). Pharmacophores can be either structure-based, by examining potential interactions between the macromolecular target and ligands, or ligand-based, by superposing an ensemble of active molecules and extracting consensus chemical features crucial for binding to a target (Yang, 2010). Typically, pharmacophores are used in virtual screening to filter large molecular databases such as PubChem (Kim et al., 2016), ChEMBL (Gaulton et al., 2012), and ZINC (Irwin & Shoichet, 2005) to identify molecules that can potentially bind to a particular target. In the virtual screening approach, pharmacophore-based filtration is usually combined with other structure-based methods such as docking and molecular dynamics, and ligand-based methods such as 2D and 3D QSARs (Yang, 2010).

In parallel, deep generative models have become central to the rational design of molecules with desired properties by learning underlying data distributions. The most widely used architectures include recurrent neural networks (RNNs), variational autoencoders (VAEs), generative adversarial networks (GANs), convolutional neural networks (CNNs), and graph neural networks (GNNs). Hence, those generative models have been trained to generate molecules with desired physicochemical properties, such as the Wildman–Crippen partition coefficient (LogP), synthetic accessibility (SA), molecular weight (MW), and quantitative estimate of drug-likeness (QED) (Pang et al., 2023). Yet, it's more challenging to design molecules with specific bioactivity profiles against therapeutic targets.

In this study, we introduce PharmaDiff, a novel generative model that produces 3D molecular structures conditioned on an atom-based representation of 3D pharmacophore hypotheses (Figure 1). By bridging the gap between pharmacophore modeling and recent advances in 3D molecular generation,

PharmaDiff aims to enable pharmacophore-informed molecule generation without requiring the 3D structure of the protein target.

## 2 RELATED WORK

Recent advances in diffusion models have enabled the equivariant generation of 3D molecular structures (Hoogeboom et al., 2022; Huang et al., 2023; Vignac et al., 2023). The Equivariant Diffusion Model (EDM) (Hoogeboom et al., 2022) was the first to use diffusion models for 3D molecular generation, employing an E(n) equivariant graph neural network (EGNN) for denoising and bond inference. Subsequent models have extended this approach with architectural innovations including transformer, graph neural networks (GNNs), and convolutional neural network (CNN)-based architectures (Alakhdar et al., 2024), most notably, MiDi (Vignac et al., 2023), which incorporates a relaxed EGNN (rEGNN) within its E(3) graph transformer layers.

Structure-based generative models such as Pocket2Mol (Peng et al., 2022), GraphBP (Liu et al., 2022), and DiffSBDD (Schneuing et al., 2024) aim to generate bioactive molecules by conditioning generation of 3D structures of the protein's active site. Some models combine that with other chemical or physical properties such as IPDiff (Huang et al., 2024) that uses protein-ligand interaction priors, and MOOD (Lee et al., 2023) that aims to generate molecules with several chemical properties such as binding affinity, drug-likeness, and synthesizability. However, those models require the 3D structure of the active site to be available either indirectly for feature extraction or directly for conditioning the model on the 3D structure. Some models, such as Mol2Mol in REINVENT 4 (Loeffler et al., 2024), constrain generation based on molecular similarity to a reference, allowing scaffold modifications; however, similarity alone may not ensure preservation of pharmacophoric features in their correct 3D arrangement.

Pharmacophore-informed generative models present an alternative approach where molecular generation is conditioned on a set of pharmacophoric features or a 3D pharmacophore hypothesis where pharmacophores act as a bridge to connect bioactivity data to generated molecules, and several models started adapting pharmacophoric features guidance in molecular generation. For example, a graph-based generative model called DEVELOP was developed to optimize both leads and hit-to-lead by designing linkers and R-groups guided by 3D pharmacophoric constraints (Imrie et al., 2021). LigDream (Skalic et al., 2019) uses a variational autoencoder (VAE) to encode the 3D representation of molecules and generate molecules conditioned on molecules' three-dimensional (3D) shape, and their pharmacophoric features. Other methods, such as TransPharmer (Xie et al., 2024) generate molecules satisfying the entire set of pharmacophore features of a reference compound but do not take into consideration the arrangement of the features in the 3D Euclidean space. PGMG (Zhu et al., 2023a) aimed to fill that gap by generating bioactive molecules against targets using the 3D pharmacophore hypothesis. However, PGMG generates molecules represented as SMILES and replaces the 3D Euclidean distance with the shortest path distances between atoms to match the pharmacophoric hypothesis (Zhu et al., 2023b).

In this work, we extend MiDi's transformer (Vignac et al., 2023) to allow the model to generate molecules that conform to predefined pharmacophoric patterns, and to achieve this goal, we introduced several changes listed in section 4.2. Moreover, by introducing a new pharmacophoric term in the loss function, our model—unlike MiDi—explicitly enforces 3D pharmacophoric constraints, ensuring that the generated molecular structures adhere to predefined pharmacophoric features without relying on protein structural data (Figure 1).

## 3 PRELIMINARIES

### 3.1 DENOISING DIFFUSION MODELS

Diffusion models are a subtype of deep generative models consisting of two Markov chains: a forward noise model and a reverse denoising model (Ho et al., 2020; Sohl-Dickstein et al., 2015). The forward diffusion model is defined by transition kernels $q(z_t|z_{t-1})$ that take input data $x_0 \sim q(x)$ and distort it by gradually injecting small amounts of noise at each time point $t \in \{1, 2, \ldots, T\}$ resulting in a trajectory of increasingly corrupted data points $(z_1, \ldots, z_T)$ such that:

$$q(z_1, \ldots, z_T | x) = q(z_1 | x) \prod_{t=2}^{T} q(z_t | z_{t-1}) \tag{1}$$

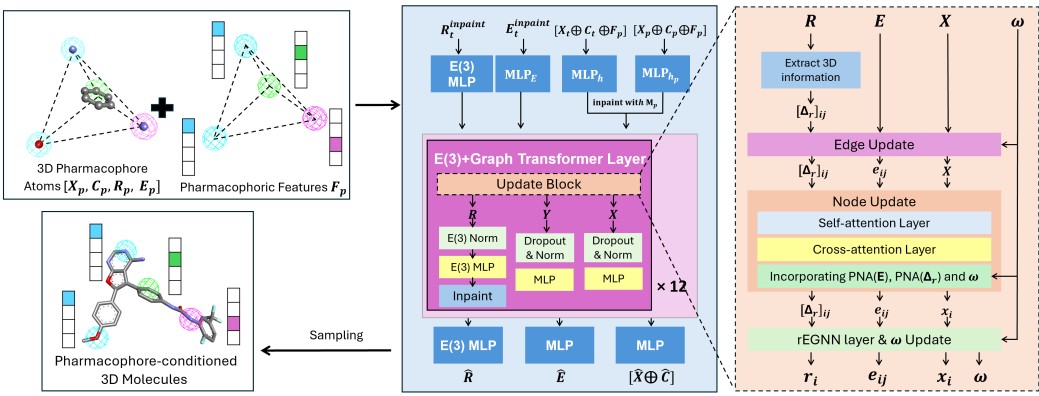

Figure 1: PharmaDiff's architecture overview simultaneously predicting 2D and 3D coordinates conditioned on a pharmacophore hypothesis. The model uses 12 layers of an E(3) graph Transformer architecture, designed to maintain SE(3) equivariance, it used inpainting and a cross-attention layer between molecular and pharmacophore Node Embeddings to enforce pharmacophore constrains.

**Gaussian diffusion for continuous data** In the case of continuous data (such as 3D atomic coordinates), Gaussian diffusion is commonly used, where noise is added at each step via a Gaussian kernel $q(z_t|z_{t-1}) \sim \mathcal{N}(\alpha_t z_t, \sigma_t^2 \mathbb{I})$. Here, $\sigma_t$ determines the amount of noise injected and $\alpha_t$ controls the amount of signal retained. Hence, we can calculate $z_t$ directly from $x$ as $q(z_t|x) \sim \mathcal{N}(\bar{\alpha}_t x, \bar{\sigma}_t^2 \mathbb{I})$ where $\bar{\alpha}_t := \prod_{s=1}^{t} \alpha_s$ and $\bar{\sigma}_t^2 = \sigma_t^2 - \alpha_t^2$.

The reverse process is modeled by a neural network $\phi_\theta$, which takes as input the noisy data $z_t$, where $z_t = \bar{\alpha}_t x + \bar{\sigma}_t \cdot \epsilon$ with $\epsilon \sim \mathcal{N}(0, \mathbb{I})$, and is trained to predict either the original clean data $x$ or the noise $\epsilon$ that was added, rather than predicting $z_{t-1}$ directly. This approach is commonly used in diffusion models (Ho et al., 2020; Sohl-Dickstein et al., 2015; Song & Ermon, 2019), because in that case the predicted data is independent of the sampled diffusion trajectory. In the case of Gaussian diffusion, the sequence trajectories are predicted as:

$$q(z_{t-1}|z_t) = \mathcal{N}(z_{t-1}; \mu_\theta(z_t, t)x, \Sigma_\theta(z_t, t)) \tag{2}$$

where $\theta$ denotes model parameters, and the mean $\mu_\theta(\mathrm{x}_t, t)$ and variance $\Sigma_\theta(\mathrm{x}_t, t)$ are parameterized by deep neural networks.

**Discrete diffusion for categorical data** Several studies (Austin et al., 2021; Vignac et al., 2023) suggest that discrete state-space diffusion is more suitable for categorical data (e.g., atom types or chemical bond types) where each state $z_t$ is a one-hot encoded vector representing categorical distribution over the $d$ possible classes (e.g., atom classes), and the forward diffusion gradually perturbs the discrete labels using a transition matrix $\mathbf{Q_t} \in R^{d \times d}$ defining the transition probabilities between discrete states. The transition kernel becomes $q(z_t|z_{t-1}) \sim \mathcal{C}(z_{t-1}\mathbf{Q_t})$ and therefore $z_t$ can be calculated from $x$ as $q(z_t = j|x) = [x\bar{\mathbf{Q}}_t]_j$ where $\bar{\mathbf{Q}}_t = \mathbf{Q_1 Q_2} \dots \mathbf{Q_t}$. In the reverse direction, the model predicts the sequence trajectories as:

$$q(z_{t-1}|z_t) \propto \sum_x p_\theta(x)(z_t \mathbf{Q'_t} \odot x\bar{\mathbf{Q}}_{t-1}) \tag{3}$$

where $\bar{\mathbf{Q}}_t = \mathbf{Q_1 Q_2} \dots \mathbf{Q_t}$, $\mathbf{Q}'$ denotes the transpose, and $\odot$ is a pointwise product.

### 3.2 Diffusion for molecules

**Molecular representation** A molecule is represented as a graph $G = (\mathbf{X}, \mathbf{C}, \mathbf{R}, \mathbf{E})$, where $\mathbf{X} \in R^{n \times d}$ is a discrete vector containing the atom type of $n$ atoms belonging to $d$ classes, $\mathbf{C} \in R^{n \times a}$ is also a one-hot encoded vector containing formal charges associated to $n$ atoms over $a$ classes. $\mathbf{R} \in R^{n \times 3}$ represents the coordinate vectors of the $n$ atoms, and $\mathbf{E} \in R^{n \times n \times b}$ contains the adjacency matrix of the one-hot bond types encoded in the $b$ classes of bond types.

**Noise model** We follow the same noise model as MiDi (Vignac et al., 2023) where Gaussian noise within the zero center-of-mass (CoM) is applied to the atomic coordinates satisfying $\sum_{i=1}^{n} \epsilon_i = 0$, which ensures roto-translation equivariance. Discrete diffusion is employed for all the discrete variables, including atom types, formal charges, and bond types. Hence, the noise model can be defined as shown in Eq.4.

$$q(G^t|G^{t-1}) \sim \mathcal{N}^{\text{CoM}}(\alpha^t \mathbf{R}^{t-1}, (\sigma^t)^2 \mathbb{I}) \times \mathcal{C}(\mathbf{X}^{t-1}\mathbf{Q}_x^t) \times \mathcal{C}(\mathbf{C}^{t-1}\mathbf{Q}_c^t) \times \mathcal{C}(\mathbf{E}^{t-1}\mathbf{Q}_e^t) \quad (4)$$

We also adapt the same adaptive noise schedules as in MiDi (Vignac et al., 2023), where the noise scheduler is manipulated, allowing for atom coordinates and bond types to be generated earlier during sampling, while atom types and formal charges are updated later in the process.

## 4 METHODS

### 4.1 FEATURIZATION OF PHARMACOPHORES

A pharmacophore hypothesis $G_p$ consists of chemical features and their spatial distribution in 3D space. Pharmacophoric features were represented with $n$ one-hot encoded discrete vectors $\mathbf{F_P} \in R^{n \times e}$, containing the feature type associated with each atom over $e$ classes of feature types. Moreover, each chemical feature is associated with one or more atoms from the molecule. For example, a hydrogen bond donor feature is associated with a single oxygen or nitrogen atom responsible for the interaction, and an aromatic feature is associated with the atoms of the aromatic ring (e.g., the six carbons of a benzene ring) and the bonds between them without any information about neighboring atoms, as shown in Figure 2.A. To help the model learn the association between the chemical features and those atoms, node features (atom types, $\mathbf{X_p}$ and charges $\mathbf{C_p}$), edge features $\mathbf{E_P}$ and positions $\mathbf{R_p}$ of atoms associated with the selected pharmacophoric features are stored as a sub-molecular pharmacophore graph $G_p = (\mathbf{X_p}, \mathbf{C_p}, \mathbf{F_p}, \mathbf{R_p}, \mathbf{E_p})$, and given as an input to the model along with the one-hot encoded pharmacophore features. Hence, the model can infer the spatial arrangement of the pharmacophoric features from the atomic positions in the pharmacophore graph.

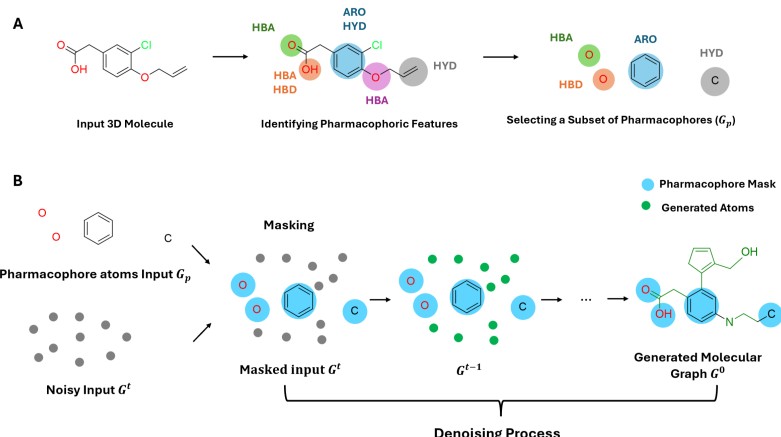

Figure 2: (A) Decomposition of molecular structures into 3D pharmacophore-associated atoms. (B) Conditioning the PharmaDiff model during the denoising process using the pharmacophore graph $G_p$. The pharmacophoric features shown include hydrogen bond acceptor (HBA), hydrogen bond donor (HBD), hydrophobic (HYD), and aromatic (ARO) groups.

### 4.2 MODEL OVERVIEW

PharmaDiff is a pharmacophore-conditioned, SE(3)-equivariant transformer-based diffusion model for 3D molecular generation. It builds upon the MiDi architecture (Vignac et al., 2023), which consists of 12 SE(3)-equivariant Graph Transformer (E3-GT) layers. Each E3-GT layer includes an encoding

multilayer perceptron (MLP), an update block with a self-attention module, followed by dropout, normalization layers, and decoding MLPs. PharmaDiff extends MiDi's transformer by conditioning the generative process on a pharmacophoric graph $G_p$, allowing the model to generate molecules that conform to predefined pharmacophoric patterns. This is achieved by several strategies to effectively integrate pharmacophoric information into the generation process, which will be described in this section. [1]

**Pharmacophore feature encoding**  To incorporate pharmacophoric information, the input molecular graph is augmented with pharmacophoric features $\mathbf{F_p}$, which are concatenated with atom types $\mathbf{X}$ and formal charges $\mathbf{C}$ to form the composite node feature vector $\mathbf{h} = [\mathbf{X}, \mathbf{C}, \mathbf{F_p}]$. This vector is then encoded using a dedicated multilayer perceptron, denoted $\mathbf{MLP_h}$, as illustrated in Figure 1. In parallel, the pharmacophore graph $G_p$ is also provided as input. Its node feature vector is defined analogously as $\mathbf{h_p} = [\mathbf{X_p}, \mathbf{C_p}, \mathbf{F_p}]$, capturing the atom types, formal charges, and pharmacophoric descriptors associated with each pharmacophore-constrained atom. These features are encoded using a separate MLP, $\mathbf{MLP_{h_p}}$, producing pharmacophore-specific embeddings used in downstream modules such as inpainting and cross-attention.

**Inpainting**  Inpainting, also referred to as the replacement method, is employed to integrate pharmacophore-associated atoms into the noisy input. This technique is commonly used to fix atoms interacting with the protein structure, as demonstrated in structure-based models such as DiffSBDD (Schneuing et al., 2024). During the inpainting process, a predefined set of mask indices $\mathbf{M_p}$ uniquely identifies the nodes corresponding to fixed atoms associated with the pharmacophore. These nodes' features, embedded by $\mathbf{MLP_{h_p}}$, along with their 3D positions $\mathbf{R_p}$ and edge features $\mathbf{E_p}$, are selectively replaced in the noisy input as the following:

$$(\mathbf{MLP_h}^{\text{inpaint}}[i],\ \mathbf{R_t}^{\text{inpaint}}[i],\ \mathbf{E_t}^{\text{inpaint}}[i,j]) = \begin{cases} (\mathbf{MLP_{h_p}}[i],\ \mathbf{R_p}[i],\ \mathbf{E_p}[i,j]), & \text{if } i \in \mathbf{M_p} \text{ and } j \in \mathbf{M_p} \\ (\mathbf{MLP_h}[i],\ \mathbf{R_t}[i],\ \mathbf{E_t}[i,j]), & \text{otherwise} \end{cases}$$

This replacement mechanism ensures that the pharmacophoric substructure is explicitly injected at the beginning of each timestep in the diffusion process. To further reinforce the 3D positional information of the pharmacophoric atoms throughout the network, an inpainting strategy is employed to reintroduce their positions, denoted as $\mathbf{R_p}$, at the end of each transformer layer. This repeated reinforcement enables the model to more effectively integrate spatial information with edge features, resulting in a more stable and structurally coherent molecular representation.

**COM adjustment**  Using inpainting to introduce the 3D positional information of pharmacophoric atoms, $\mathbf{R_p}$, throughout the network can shift the center of mass (COM) and compromise the E(3)-equivariance. To prevent this, we realign the noisy atomic positions before inpainting by translating them such that the input's COM is centered at the origin $\sum_{i=1}^{n} r_i = 0$.

$$\tilde{\mathbf{R}}_t^{\text{input}} = \mathbf{R}_t^{\text{input}} + \frac{1}{n}\sum_{i \in \mathbf{M_p}} \mathbf{R}_p - \frac{1}{n}\sum_{i \in \mathbf{M_p}} \mathbf{R}_{t,i}^{\text{input}}$$

**Cross attention**  To facilitate a more seamless and context-aware integration of pharmacophoric constraints into the molecular representation, we introduce cross-attention layers between the generated node embeddings and the pharmacophoric node embeddings. In this setup, each generated atom node (query) attends to the pharmacophoric atom nodes (keys and values), enabling the molecular representation to be dynamically influenced by the most relevant pharmacophoric features. Compared to static approaches like inpainting, this method provides a more flexible and learnable fusion of pharmacophoric information into the molecule. Formally, let $\mathbf{H}_{\text{mol}} \in \mathbb{R}^{N \times d}$ denote the molecular node embeddings and $\mathbf{H}_{\text{pharm}} \in \mathbb{R}^{M \times d}$ denote the pharmacophoric node embeddings. The cross-attention projections are computed as follows:

$$\mathbf{Q} = \mathbf{H}_{\text{mol}}\mathbf{W}_Q, \quad \mathbf{K} = \mathbf{H}_{\text{pharm}}\mathbf{W}_K, \quad \mathbf{V} = \mathbf{H}_{\text{pharm}}\mathbf{W}_V$$

where $\mathbf{W}_Q, \mathbf{W}_K, \mathbf{W}_V \in \mathbb{R}^{d \times d}$ are learnable projection matrices for the **query**, **key**, and **value**, respectively. The cross-attention output is then computed as:

---

[1]The software developed for this study and the processed datasets will be made publicly available on GitHub after the peer-review process. A temporary anonymous version is available at `https://github.com/pharmadiff/PharmaDiff`.

$$\mathbf{O} = \mathbf{W}_{\text{out}} \left( \text{Softmax} \left( \frac{\mathbf{W}_{\text{A}}(\mathbf{Q}\mathbf{K}^{\top})}{\sqrt{d}} \right) \mathbf{V} \right)$$

Here $\mathbf{W}_{\text{out}} \in \mathbb{R}^{d \times d}$ and $\mathbf{W}_{\text{A}} \in \mathbb{R}^{d \times d}$ are additional learnable projections. The output is $\mathbf{O}$ is then used to inpaint $\mathbf{H}_{\text{mol}}$.

**Loss function**  Our denoising network is trained to recover the original molecular structure $G = (\mathbf{X}, \mathbf{C}, \mathbf{R}, \mathbf{E})$ from a noisy input $G_t$. As shown in the loss function, the prediction of atomic coordinates $\hat{\mathbf{R}}$ is treated as a regression problem, optimized using mean-squared error (MSE), while the predictions for atom types ($\hat{\mathbf{X}}$), formal charges ($\hat{\mathbf{C}}$), and bond types ($\hat{\mathbf{E}}$) are optimized as classification problems, trained with cross-entropy loss (CE).

However, unlike MiDi, our model explicitly incorporates pharmacophoric constraints to ensure that the generated molecular structure adheres to the predefined pharmacophoric features. To this end, we place greater emphasis on the accurate prediction of pharmacophoric atom types $\mathbf{X_p}$, formal charges $\mathbf{C_p}$, and positions $\mathbf{R_p}$ compared to the rest of the molecular atoms. This is achieved through an additional pharmacophore loss term, which imposes specific position, atom type, and charge constraints on the atoms associated with the pharmacophore. The final loss function is formulated as:

$$l(G, \hat{G}) = \underbrace{\lambda_r ||\hat{\mathbf{R}} - \mathbf{R}||^2 + \lambda_x \text{CE}(\mathbf{X}, \hat{\mathbf{X}}) + \lambda_c \text{CE}(\mathbf{C}, \hat{\mathbf{C}}) + \lambda_e \text{CE}(\mathbf{E}, \hat{\mathbf{E}})}_{\text{Molecular generation loss}}$$
$$+ \underbrace{\lambda_{r_p} ||\hat{\mathbf{R}} - \mathbf{R_p}||^2 + \lambda_{x_p} \text{CE}(\mathbf{X_p}, \hat{\mathbf{X}}) + \lambda_{c_p} \text{CE}(\mathbf{C_p}, \hat{\mathbf{C}})}_{\text{Pharmacophore loss}}$$

Here, $\lambda_r$, $\lambda_x$, $\lambda_c$, $\lambda_e$, $\lambda_{r_p}$, $\lambda_{x_p}$, and $\lambda_{c_p}$ are weighting factors that control the relative importance of different components of the loss. Those hyperparameter were selected based in a series of short numerical experiments.

## 5 EXPERIMENTS

### 5.1 DATASET

The model was trained on the GEOM-DRUGs dataset (Axelrod & Gomez-Bombarelli, 2022), which is suitable for drug design applications given that it contains over 450,000 drug-sized molecules with an average of 44.4 atoms (24.9 heavy atoms) and up to a maximum of 181 atoms (91 heavy atoms). The dataset was split into 80% for training, 10% for validation, and 10% for testing. For each molecule, the five lowest energy conformations were selected to build the dataset, and for each conformer, all the chemical features of a molecule are identified using RDKit, and a subset of 3-7 random features was selected to build the 3D pharmacophoric hypothesis. For the structure-based drug design experiment, the structure of the targets: BRD4 (PDBID: 3MXF), VEGFR2 (PDBID: 1YWN), CDK6 (PDBID: 2EUF), and TGFB1 (PDBID: 6B8Y) were downloaded from the PDB database (Burley et al., 2017).

### 5.2 BASELINES

In the ligand-based experiment, to evaluate the effectiveness of PharmaDiff in pharmacophore-conditioned molecular generation, we randomly selected 250 compounds, referred to as conditioning compounds, along with their corresponding 3D pharmacophore hypotheses from the reserved test set of the GEOM-Drugs dataset. Each pharmacophore hypothesis consisted of 3 to 7 features, such as hydrogen bond donors, acceptors, hydrophobic groups, or aromatic rings, randomly extracted from the conditioning compounds. These hypotheses were used to guide the de novo generation of 3D molecular structures. We compare our method with TransPharmer's (72bit, 108bit, 1032bit) models (Xie et al., 2024), REINVENT 4's Mol2Mol high and medium similarity models (Loeffler et al., 2024), and PGMG (Zhu et al., 2023a). For TransPharmer and REINVENT 4 models, 100 molecules were generated per conditioning compound, each satisfying the full set of pharmacophoric features derived from the reference compound. PGMG, by contrast, directly employs a 3D pharmacophore hypothesis to guide the generation process. For all three baseline models, 3D conformations of

the generated molecules were constructed from SMILES representations using RDKit's ETKDG algorithm, followed by geometry refinement via energy minimization with the MMFF94 force field (Halgren, 1996). For this experiment, results are reported as the mean ± standard errors of per-condition values, reflecting variability across the 250 pharmacophore conditions (Tables 1 and 2). For the structure-based setting, we evaluated the ability of PharmaDiff to generate ligands compatible with the active sites of four proteins: BRD4, VEGFR2, CDK6, and TGFB1 with pharmacophore hypotheses reported in (Zhu et al., 2023a) and collected from the literature (Lee et al., 2010; Shawky et al., 2021; Jiang et al., 2018; Roskoski Jr, 2019; Yan et al., 2018). The 3D coordinates and types of pharmacophore-associated atoms were provided as fixed input. PharmaDiff was compared against DiffSBDD-cond (Schneuing et al., 2024), which uses protein structure and employs inpainting to inject information about fixed parts of the ligand into the sampling process. We provided the pharmacophore-associated atom positions and types as input for the fixed atoms. For each protein target, 1000 molecules were sampled from each model. For this experiment, results are reported as the mean ± standard errors of per-molecule values, reflecting variability across the generated molecules (Table 3). Moreover, ablation studies and other case studies were performed and are described in detail in the Appendix section A.2.

## 5.3 Evaluation metrics

**Basic generation metrics**   The quality of the generated molecules was assessed using four core metrics. **Validity** was measured by the success rate of RDKit sanitization. **Uniqueness** represents the proportion of valid molecules that have distinct canonical SMILES. **Novelty** is defined as the fraction of unique molecules whose canonical SMILES do not appear in the training set. Finally, **Diversity** was calculated as $1 -$ average pairwise Tanimoto similarity (Tanimoto, 1958; Willett et al., 1998), using Morgan fingerprints (Rogers & Hahn, 2010) (radius 2, 2048 bits). Diversity values range from 0 to 1, with the reported value representing the average across all generation conditions.

**Pharmacophore match evaluation**   To assess how well the generated molecules match the pharmacophore hypotheses they were conditioned on, we used two key metrics. The **Match Score (MS)**, adapted from PGMG (Zhu et al., 2023a), quantifies the degree to which molecules align with their 3D pharmacophoric hypotheses, and it ranges from 0 to 1, where an MS of 1.0 entails a perfect match. Unlike the original PGMG implementation, which uses shortest path (graph-based) distances, we computed MS using actual Euclidean distances in 3D space between atoms. The second metric, **Perfect Match Rate (PMR)**, reflects the proportion of generated molecules that achieved an MS of 1, corresponding to a perfect match ($MS = 1$) to the specified 3D pharmacophore hypothesis.

**Binding affinity estimation**   We estimated the binding affinity of generated molecules using the **Vina Score** (Trott & Olson, 2010), a scoring function from AutoDock Vina that quantifies the predicted interaction strength between ligands and protein targets in molecular docking simulations. Average Vina scores are reported for the top 1, 100, and all of the generated molecules.

**Drug-likeness and physicochemical properties**   To evaluate the potential bioactivity of the generated molecules, we also report several key chemical properties of the generated molecules that are particularly relevant for drug design applications. These include molecular weight, quantitative estimate of drug-likeness (QED), synthetic accessibility (SA) score (Ertl & Schuffenhauer, 2009), partition coefficient (logP), topological polar surface area (TPSA), and the average number of rings.

## 5.4 Results

### 5.4.1 3D Pharmacophore matching for ligand-based drug design

In terms of standard molecular generation evaluation metrics, as shown in Table 1, PharmaDiff achieves the highest novelty (0.9989) and the second-highest uniqueness (0.9933) just after PGMG, with only a minor difference. Although PharmaDiff's validity rate of 0.8823 is slightly lower than the SMILES-based generators PGMG, TransPharmer, and REINVENT4 (which often exceed 0.94), it is on par with other 3D approaches, especially models using molecular inpainting or fragment-based design (Igashov et al., 2022; Schneuing et al., 2024). Notably, PharmaDiff excels in diversity (0.8686), outperforming all other models and indicating a broader exploration of the chemical space.

In terms of molecular-pharmacophore alignment metrics (Table 2), results reveal that PharmaDiff achieves the highest MS average (0.8964), along with the highest fraction of molecules matching exactly to the pharmacophore with an average **PMR** of 0.6990. It also ranks well with 80.50% of generated molecules having an $MS \geq 0.8$. While the average matching scores (MS) across models vary only slightly, particularly when compared to similarity-driven methods such as REINVENT4 or the general pharmacophore-matching approach used in TransPharmer, the proportion of molecules that either exactly match the pharmacophore hypothesis (PMR) or closely align with it (%MS $\geq 0.8$) is significantly higher in models explicitly conditioned on a specific 3D pharmacophore. Notably, PharmaDiff achieves superior performance compared to PGMG, likely due to its reliance on true 3D Euclidean distances for pharmacophore matching, as opposed to the graph-based shortest path approximations employed by PGMG. Examples of the generated molecules are shown in Figure 6 in the appendix.

Table 1: General generation performance of pharmacophore-conditioned molecular generative models, results are reported as mean ± error across all 250 conditions.

| Metric | Validity ↑ | Uniqueness ↑ | Novelty ↑ | Diversity ↑ |
|---|---|---|---|---|
| PharmaDiff (3D) | 0.8823 ± 0.0037 | 0.9933 ± 0.0020 | **0.9989 ± 0.0002** | **0.8686 ± 0.0028** |
| PGMG (3D) + MMFF | 0.9439 ± 0.0039 | **0.9945 ± 0.0007** | 0.9945 ± 0.0010 | 0.8294 ± 0.0019 |
| TransPharmer-72bit (3D) + MMFF | **0.9922 ± 0.0013** | 0.8928 ± 0.0074 | 0.9556 ± 0.0034 | 0.7503 ± 0.0049 |
| TransPharmer-108bit (3D) + MMFF | 0.9900 ± 0.0018 | 0.7658 ± 0.0116 | 0.9628 ± 0.0032 | 0.6702 ± 0.0074 |
| TransPharmer-1032bit (3D) + MMFF | 0.9690 ± 0.0036 | 0.7175 ± 0.0130 | 0.9790 ± 0.0021 | 0.6148 ± 0.0085 |
| REINVENT4 medium similarity (3D) + MMFF | 0.9710 ± 0.0068 | 0.9877 ± 0.0015 | 0.9797 ± 0.0038 | 0.5282 ± 0.0036 |
| REINVENT4 high similarity (3D) + MMFF | 0.9551 ± 0.0081 | 0.9755 ± 0.0021 | 0.9960 ± 0.0016 | 0.4210 ± 0.0034 |

Table 2: Pharmacophore matching performance of pharmacophore-conditioned molecular generative models.

| Metric | MS ↑ | PMR ↑ | %$MS \geq 0.8$ ↑ |
|---|---|---|---|
| PharmaDiff (3D) | **0.8964 ± 0.0050** | **0.6990 ± 0.0142** | 0.8050 ± 0.0094 |
| PGMG (3D) + MMFF | 0.8938 ± 0.0063 | 0.5099 ± 0.0225 | **0.8286 ± 0.0131** |
| TransPharmer-72bit (3D) + MMFF | 0.8125 ± 0.0077 | 0.2795 ± 0.0204 | 0.6383 ± 0.0195 |
| TransPharmer-108bit (3D) + MMFF | 0.8233 ± 0.0078 | 0.3171 ± 0.0218 | 0.6570 ± 0.0199 |
| TransPharmer-1032bit (3D) + MMFF | 0.8313 ± 0.0078 | 0.3477 ± 0.0225 | 0.6758 ± 0.0203 |
| REINVENT4 medium similarity (3D) + MMFF | 0.8424 ± 0.0080 | 0.3826 ± 0.0234 | 0.7045 ± 0.0196 |
| REINVENT4 high similarity (3D) + MMFF | 0.8664 ± 0.0072 | 0.4252 ± 0.0240 | 0.7530 ± 0.0189 |

To assess the relevance of the generated molecules as potential drugs, we compared their physico-chemical property distributions to those of the training data. As illustrated in Fig. 3, the generated molecules closely resemble the GEOM-Drugs dataset in terms of key properties, including molecular weight (MW), LogP, QED, TPSA, and ring count. While PharmaDiff exhibits slightly higher synthetic accessibility (SA) scores, it remains well within the acceptable range for drug-like compounds. These results suggest that PharmaDiff effectively captures and reproduces the molecular property distribution of its training data.

### 5.4.2 DEMONSTRATION OF APPLICATION IN STRUCTURE-BASED DRUG DESIGN

Table 3 compares PharmaDiff and DiffSBDD across four protein targets: 1ywn (6 features), 2euf (3 features), 3mxf (4 features), and 6b8y (4 features), using docking scores, synthetic accessibility, diversity, and pharmacophore matching metrics. PharmaDiff consistently achieves better average Vina scores across all targets, and stronger top-100 docking performance in three out of the four targets, particularly excelling on 1ywn and 6b8y, which have more complex pharmacophore constraints. While Top-1 scores are more competitive, with DiffSBDD slightly outperforming on 3mxf and 6b8y, PharmaDiff still matches or exceeds performance on the remaining targets. PharmaDiff also demonstrates better synthetic feasibility, as shown by consistently lower SA scores across all targets, indicating its ability to generate realistic and accessible molecules even under stricter pharmacophore demands, such as those in 1ywn. In terms of chemical diversity, DiffSBDD holds a slight advantage across the board; however, the difference is minor, and PharmaDiff maintains high diversity while also enforcing complex constraints. Most notably, PharmaDiff significantly outperforms DiffSBDD in pharmacophore matching score (MS) across all targets, especially in 1ywn and 2euf, suggesting a higher fidelity in satisfying both simple (3 features) and more intricate (6 features) 3D pharmacophore patterns during molecule generation.

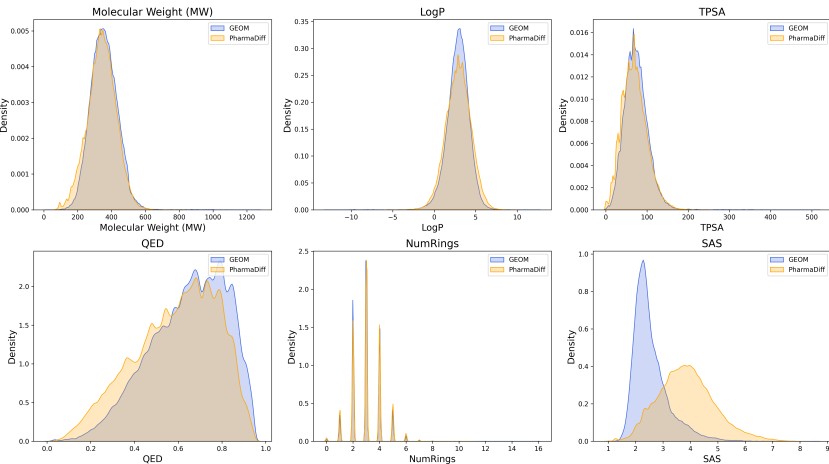

Figure 3: Distribution of physicochemical properties for the GEOM-Drugs training set and molecules generated by PharmaDiff. The plot compares 100,000 molecules generated from random pharmacophore hypotheses with 100,000 conformers randomly selected from the GEOM-Drugs training set.

Table 3: Comparison of docking scores and drug-like properties between PharmaDiff and DiffSBDD across four protein targets, results are reported as mean ± standard error (SE)

| Protein | Method | Vina score (Min) ↓ All (kcal/mol) | Vina score (Min) ↓ Top 1 (kcal/mol) | Vina score (Min) ↓ Top 100 (kcal/mol) | SA Score ↓ Top 100 | Diversity ↑ Top 100 | MS ↑ Top 100 |
|---|---|---|---|---|---|---|---|
| 1ywn | PharmaDiff | **-5.7730 ± 0.0513** | **-10.6380** | **-8.3185 ± 0.0489** | **3.3236 ± 0.0560** | 0.8098 ± 0.0008 | **0.8840 ± 0.0150** |
| | DiffSBDD | -4.6341 ± 0.0224 | -8.9850 | -6.4088 ± 0.1175 | 3.6607 ± 0.0961 | **0.8373 ± 0.0013** | 0.5827 ± 0.0306 |
| 2euf | PharmaDiff | **-7.4273 ± 0.0405** | **-9.2768** | **-9.2767 ± 0.0388** | **3.2241 ± 0.0657** | 0.8309 ± 0.0008 | **0.9700 ± 0.0126** |
| | DiffSBDD | -7.2906 ± 0.0379 | -9.0236 | -9.0236 ± 0.0435 | 3.8148 ± 0.0781 | **0.8737 ± 0.0005** | 0.7833 ± 0.0290 |
| 3mxf | PharmaDiff | **-6.8463 ± 0.0355** | -10.7130 | -8.5968 ± 0.0485 | **3.5691 ± 0.0642** | 0.8537 ± 0.0006 | **0.8183 ± 0.0203** |
| | DiffSBDD | -5.7141 ± 0.0535 | **-10.8790** | **-8.6539 ± 0.0640** | 4.2065 ± 0.0673 | **0.8830 ± 0.0005** | 0.5767 ± 0.0236 |
| 6b8y | PharmaDiff | **-7.8427 ± 0.0440** | -11.4270 | **-9.7503 ± 0.0485** | **3.4352 ± 0.0500** | 0.8307 ± 0.0008 | **0.8800 ± 0.0126** |
| | DiffSBDD | -6.2078 ± 0.0601 | **-12.1900** | -9.4052 ± 0.0706 | 3.7569 ± 0.0800 | **0.8818 ± 0.0006** | 0.7217 ± 0.0250 |

# 6 Discussions and Conclusions

PharmaDiff demonstrates strong performance in pharmacophore-conditioned generation—achieving higher pharmacophore match accuracy than SMILES-based baselines and outperforming prior 3D structure-based generative models in docking scores. However, the model still faces challenges, particularly in the inpainting process where atoms are modified to align with predefined pharmacophoric features. While this approach can retain important properties such as aromaticity or the presence of hydrogen bond acceptors, it may occasionally lose key pharmacophoric features when the local atomic context changes. Subtle structural modifications can disrupt hydrogen bonding, hydrophobic interactions, or other non-covalent forces essential for target binding and biological activity. Although PharmaDiff implicitly incorporates pharmacophoric feature types as contextual input during generation, these features are not explicitly reinforced. Despite the strong performance achieved through this implicit integration—evidenced by an average PMR of 0.6990—future work could explore explicitly enforcing the retention of pharmacophoric features in the loss function through a differentiable, feature-aware model to better preserve those critical for target interactions. An additional challenge common to atom-level inpainting approaches—including PharmaDiff and similar models (Runcie & Mey, 2023; Schneuing et al., 2024)—is the generation of disconnected molecular structures, particularly when handling complex pharmacophores with several features and thus a large number of associated inpainted atoms. The common practice of retaining only the largest connected fragment to maintain connectivity can lead to the exclusion of pharmacophore-associated atoms, potentially resulting in the loss of critical functional groups. As a result, the generated molecules may lack features essential for bioactivity. Future improvements could focus on enhancing connectivity within the inpainting process through stricter structural constraints, energy-guided sampling, or post-processing optimization and energy-minimization techniques that preserve both the overall molecular integrity and key pharmacophoric elements.

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

# A APPENDIX

## A.1 IMPLEMENTATION DETAILS

### A.1.1 SOFTWARE

Dataset processing was done in Python (v.3.9.) using RDKit (2022.03.2) for molecular preprocessing, chemical structure handling, pharmacophore features extraction, MMFF minimization and calculating SA, QED and molecular fingerprints. PyTorch (v2.0.1, CUDA 11.8), PyTorch Lightning (v2.0.4), and Torch Geometric (v2.3.1) were used for model development and training. Configuration was managed with Hydra (v1.3.2) and OmegaConf (v2.3.0). Logging and experiment tracking were handled via Weights & Biases (wandb, v0.15.4).

Baseline methods were obtained from their official implementations with only minimal adaptations for input/output compatibility. PGMG Zhu et al. (2023a) was used under the CC BY-NC-SA 4.0 License from https://github.com/CSUBioGroup/PGMG. REINVENT4 Loeffler et al. (2024) (Apache 2.0) was obtained from https://github.com/MolecularAI/REINVENT4. TransPharmer Xie et al. (2024) and DiffSBDD Schneuing et al. (2024) were both used under the MIT License from https://github.com/iipharma/transpharmer-repo and https://github.com/arneschneuing/DiffSBDD, respectively.

### A.1.2 Hardware

Data processing was conducted on an RM-512 partition of a high-performance computing (HPC) system, Model training was performed on an NVIDIA RTX 3090 Ti GPU over 390 epochs, requiring a total of 16 days. The 25k molecules sampled, with 100 molecules for each of the 250 conditions, took around one day.

In addition, the baseline or comparison method was executed on an HPC system equipped with NVIDIA V100 GPU nodes. Molecular structures were converted from SMILES to 3D conformers, followed by MMFF (Merck Molecular Force Field) minimization, all of which were executed on RM nodes of a high-performance computing (HPC) system.

### A.2 Additional Analyses

#### A.2.1 Ablation study: impact of inpainting and cross attention on pharmacophore-conditioned generation

To evaluate the role of inpainting in PharmaDiff's pharmacophore-constrained molecular generation, we performed an ablation study comparing the full PharmaDiff model against a variant with inpainting disabled and a variant with the cross-attention mechanism disabled. In the full model, inpainting provides explicit spatial and chemical control by conditioning the generation on both the 3D coordinates and identities of pharmacophore-associated atoms/fragments. In the ablated version, this detailed conditioning was removed. Only the pharmacophore types (e.g., donor, acceptor, hydrophobe) were provided, without positional or fragment-level information, effectively eliminating spatial pharmacophore guidance.

These results (as shown in Table 4) clearly highlight how important inpainting is for generating high-quality molecules satisfying the pharmacophore hypothesis. Validity dropped from 0.8789 to 0.6494 without inpainting, suggesting a decline in structural coherence. While uniqueness and novelty remained high in both settings, pharmacophore match scores decreased significantly: the average MS declined from 0.8995 to 0.6899, and the proportion of molecules fully satisfying the pharmacophore (**PMR**) dropped from 0.6823 to just 0.2105. For cross attention, disabling it led to a minor decrease in the validity, dropping to 0.8682, and a more notable decline in MS to 0.8335, and **PMR** to 0.6423. These findings demonstrate that inpainting is essential for ensuring both feature perseverance and spatial alignment in pharmacophore-conditioned generation, while cross attention plays a minor role. Without inpainting, the model struggles to translate abstract feature types into correctly positioned and chemically appropriate substructures.

Table 4: Ablation study comparing PharmaDiff with and without inpainting. Results are reported over 1,024 generated molecules.

| Ablation Study: Effect of Inpainting and cross attention | | | | | |
|---|---|---|---|---|---|
| Model | Validity ↑ | Uniqueness ↑ | Novelty ↑ | MS ↑ | PMR ↑ |
| PharmaDiff (3D) | **0.8789** | 0.9900 | 1.00 | **0.8995** | **0.6823** |
| PharmaDiff (3D, no inpainting) | 0.6494 | **1.00** | 1.00 | 0.6899 | 0.2105 |
| PharmaDiff (no cross attention) | 0.8682 | 0.9933 | 1.00 | 0.8335 | 0.6423 |

#### A.2.2 Case study: design of BBB-permeable PI3Kα inhibitors via pharmacophore-guided generation

Phosphoinositide 3-kinase alpha (PI3K$\alpha$) plays a central role in oncogenic signaling and is frequently activated in solid tumors such as breast, colorectal, and brain cancers. Its dysregulation promotes tumor proliferation, survival, and resistance to targeted therapies. Despite the development of several PI3K$\alpha$ inhibitors, their effectiveness against brain cancers remains limited due to poor penetration of the blood–brain barrier (BBB). To overcome this limitation, we implemented a scaffold hopping strategy aimed at designing PI3K$\alpha$ inhibitors with enhanced central nervous system (CNS) drug-like properties Lian et al. (2024). A structure-based pharmacophore model was derived from the co-crystal structure of *Taselisib* bound to PI3K$\alpha$ (PDB ID: 8EXL) using PLIP Adasme et al. (2021). Guided by this pharmacophore hypothesis, we generated a library of 5,000 molecules through scaffold replacement, retaining the essential interaction-driving pharmacophore features while introducing scaffold diversity. In particular, we aimed to replace *Taselisib*'s hydrophilic heterocycles, specifically

the pyrazole and the triazole rings, which contribute significantly to its topological polar surface area (TPSA) and may hinder BBB permeability, with less polar scaffolds that preserve the spatial arrangement of key pharmacophoric groups. As illustrated in Figure 4, PharmaDiff successfully generated molecules with chemically diverse scaffolds that preserved the 3D spatial alignment of the pharmacophore hypothesis with the reference ligand, *Taselisib*.

To prioritize BBB-permeant candidates, we applied CNS drug-like property filters based on molecular weight ($\leq 400$ Da), logP (2–4), topological polar surface area (TPSA $\leq 90$ Å$^2$), hydrogen bond donors ($\leq 2$), and acceptors ($\leq 6$). Molecular descriptors were computed using RDKit. This filtering step yielded 441 compounds, 433 of which were unique. All filtered compounds were docked to the PI3K$\alpha$ structure (8EXL) using AutoDock Vina, focusing on the *Taselisib* binding site. The best-scoring pose for each molecule was retained for further analysis. Of these, 113 compounds (26.10%) achieved docking scores below $-8.0$ kcal/mol, and **39 compounds (9.01%)** surpassed the reference ligand *Taselisib*, which scored $-8.512$ kcal/mol. These results demonstrate the potential of pharmacophore-guided scaffold hopping to identify novel PI3K$\alpha$ inhibitors with improved BBB permeability and potent binding affinity. As illustrated in Figure 4, PharmaDiff was able to generate diverse scaffolds, allowing us to explore alternative core structures while preserving key pharmacophoric interactions essential for PI3K$\alpha$ binding.

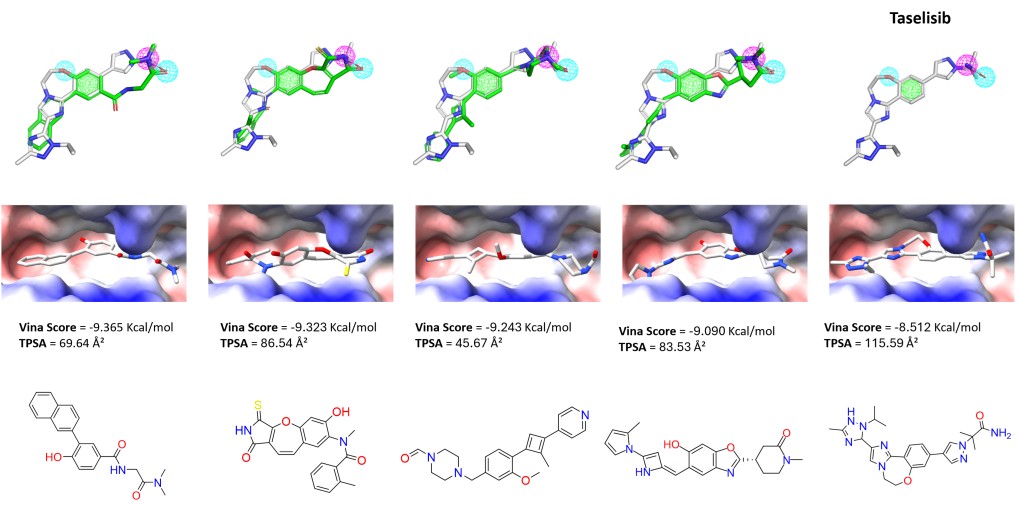

Figure 4: Top 4 molecules ranked by docking score, showing chemically diverse scaffolds generated by PharmaDiff. The top row illustrates the spatial alignment of pharmacophoric features in the generated ligands (green) to the pharmacophore derived from *Taselisib* (grey). Pharmacophore features are color-coded as follows: hydrogen bond acceptors in cyan, hydrophobes in green, and hydrogen bond donors in magenta. The middle row shows the docked poses of the generated ligands within the PI3K$\alpha$ binding pocket. The bottom row displays the corresponding 2D chemical structures, highlighting scaffold diversity among the top-scoring candidates.

### A.2.3 Case study: pharmacophore-Guided R-Group Sampling in CDK2 inhibitors

Cyclin-dependent kinase 2 (CDK2) regulates the G1/S phase transition of the cell cycle and is frequently dysregulated in a variety of cancers, making it a well-studied therapeutic target. *Roscovitine*, a purine-based ATP-competitive inhibitor, binds to CDK2 as captured in the co-crystal structure PDBID: 2A4L Azevedo et al. (1997). By inspecting this structure with PLIP Adasme et al. (2021), we extracted a set of pharmacophore features that reflect key ligand–protein interactions. To test the ability of PharmaDiff to generate analogs that preserve these interactions while allowing chemical variation in the R substituents, we followed a structured protocol. We first extracted 3D pharma-

cophore points from the ligand–protein complex and then sampled small molecular fragments or atoms corresponding to each pharmacophore type, such as oxygen and nitrogen atoms for hydrogen bond donors and acceptors, and aromatic heterocycles like pyridine, pyrimidine, imidazole, or furan for the aromatic site. A variety of hydrophobic fragments were also incorporated, including benzene rings, cycloalkanes (e.g., cyclobutane, cyclopropane), and branched groups such as isopropyl or tert-butyl. The 3D geometries of all fragments were minimized using the MMFF94 force field. Finally, PharmaDiff was conditioned on the combined pharmacophore and its associated 3D atoms/fragment inputs to generate a set of 200 candidate molecules.

As shown in Figure 5, the generated molecules align well with the active conformation of *Roscovitine* in the CDK2 binding pocket. Remarkably, this approach for R group sampling enables PharmaDiff to generate molecules with various chemical groups under the same pharmacophore constraints. For example, the aromatic ring is fulfilled by a variety of heteroaromatics, including pyridine, pyrimidine, furan and imidazole, while the hydrophobic pharmacophore is matched by structurally diverse fragments such as branched alkyl groups, benzene and cycloalkyl rings. These findings highlight PharmaDiff's potential for fragment replacement, and R-group exploration in lead optimization workflows.

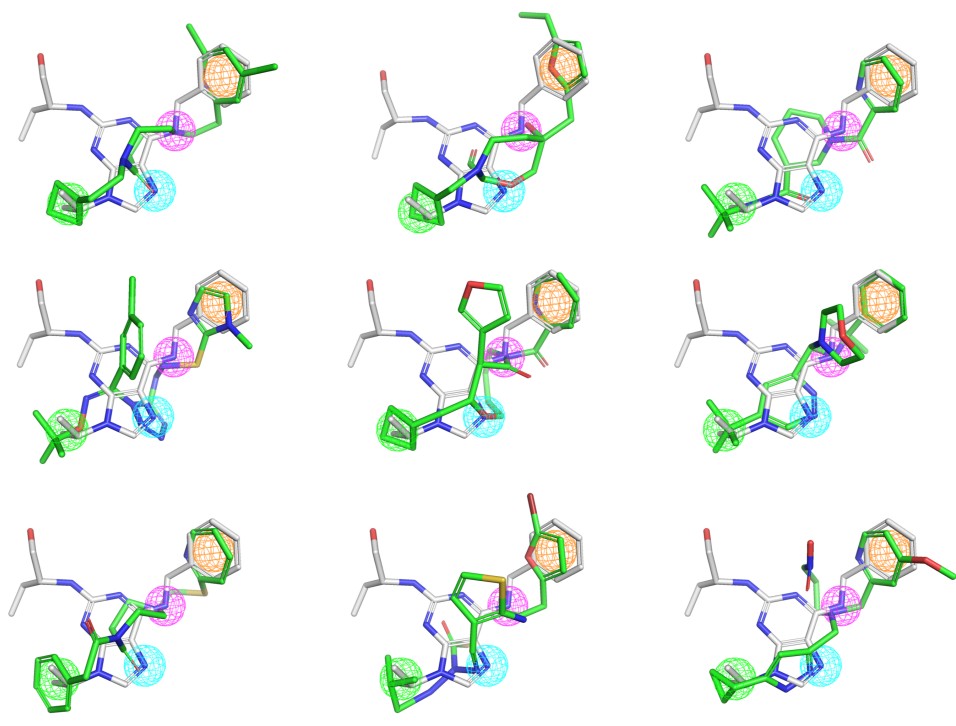

Figure 5: pharmacophore-aligned generated ligands (green) compared to the reference CDK2 inhibitor ligand *Roscovitine* (grey). Pharmacophore features are color-coded as follows: hydrogen bond acceptors in cyan, hydrophobes in green, and aromatic groups in orange and hydrogen bond donors in magenta.

LARGE LANGUAGE MODEL (LLM) DISCLOSURE

The authors used a large language model (ChatGPT, OpenAI, 2025) to aid in polishing the writing and formatting of text (e.g., LaTeX syntax, grammar, and figure captions). All scientific content, results, and conclusions are entirely the work of the authors.

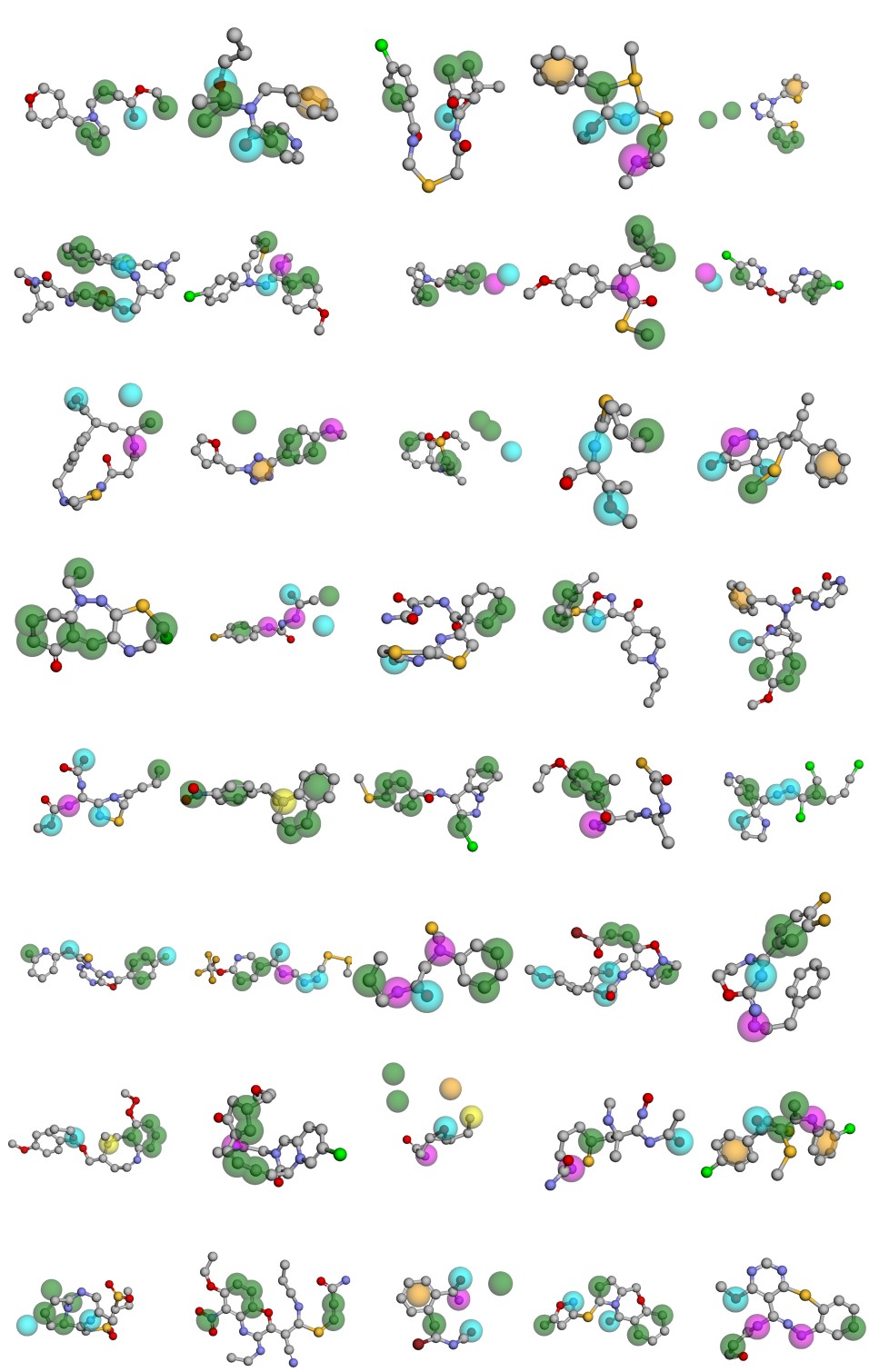

Figure 6: Examples of non-curated molecules generated conditionally on a predefined pharmacophore hypothesis. Molecules are shown as stick representations, while pharmacophore features are visualized as spheres: hydrophobic (green), hydrogen-bond acceptors (cyan), hydrogen-bond donors (magenta), aromatic features (orange), and positively ionizable features (yellow).

