# OpenReview forum: "Pharmacophore-Conditioned Diffusion Model for Ligand-Based De Novo Drug Design"
_ICLR.cc/2026/Conference — ICLR 2026 Conference Withdrawn Submission_

### Official Review · Reviewer_ojg9 · 2025-10-20

**Soundness:** 2
**Presentation:** 2
**Contribution:** 2
**Rating:** 2
**Confidence:** 4

**Summary:**

This work presents PharmaDiff, a pharmacophore-conditioned diffusion model for 3D molecular generation that enables ligand-based de novo drug design without the need for target protein structures. The approach embeds pharmacophoric constraints into an SE(3)-equivariant transformer diffusion framework, enhancing the MiDi architecture through the introduction of inpainting, cross-attention modules, and pharmacophore-aware loss functions.


Experimental results on the GEOM-Drugs dataset show that PharmaDiff achieves superior pharmacophore alignment (MS = 0.8964, PMR = 0.6990) and competitive docking performance across multiple protein targets compared to PGMG, TransPharmer, and DiffSBDD, while preserving strong chemical diversity and synthetic accessibility.

**Strengths:**

* This paper introduces pharmacophore conditioned generation for molecular design, which is crucial when analyzing interaction patterns between drugs and targets.
* PharmaDiff shows improvements in terms of pharmacophore matching and competetive docking performance compared to DiffSBDD.

**Weaknesses:**

* One primary concern of this work is the relationship between 'pharmacophore' and 'ligand-based drug deisgn'. Pharmacophore contains information like hydrogen bond donor (HBD), hydrogen bond acceptor (HBA), etc., which are important parts when forming H-bonds between ligands and targets. However, this paper claims a setting where target protein is not needed. In such a setting, the pharmacophore becomes an abstract spatial pattern inferred solely from ligand conformations rather than a true interaction map. This raises questions about whether the conditioning genuinely captures biologically meaningful interaction constraints or merely enforces geometric regularities observed among similar ligands.
* The experimental section appears relatively weak and does not convincingly demonstrate the claimed advantages of PharmaDiff. First, the reported improvement in pharmacophore matching is modest and does not clearly surpass strong ligand-based baselines by a significant margin, raising questions about the practical impact of the proposed conditioning strategy. Second, in the structure-based drug design (SBDD) experiments, the comparison is incomplete—many relevant baselines are omitted[1-3], and the evaluation covers only four protein targets, which limits the generality of the conclusions. Finally, an important recent baseline, Shepherd[4], which explicitly incorporates pharmacophore-based molecular design, is not included or discussed. This omission makes it difficult to assess whether PharmaDiff provides meaningful advances over state-of-the-art pharmacophore-conditioned generative models. A discussion or comparison with Shepherd would be necessary to fairly position the contribution of this work.
* While the pharmacophore-conditioning is interesting, the generative backbone (E(3)-GT + diffusion) largely extends MiDi with additional conditioning terms. The conceptual leap might be seen as incremental rather than foundational.


[1] Guan J, Qian W W, Peng X, et al. 3d equivariant diffusion for target-aware molecule generation and affinity prediction[J]. arXiv preprint arXiv:2303.03543, 2023.

[2] Pinheiro P O, Jamasb A, Mahmood O, et al. Structure-based drug design by denoising voxel grids[J]. arXiv preprint arXiv:2405.03961, 2024.

[3] Qu Y, Qiu K, Song Y, et al. MolCRAFT: structure-based drug design in continuous parameter space[J]. arXiv preprint arXiv:2404.12141, 2024.

[4] Adams K, Abeywardane K, Fromer J C, et al. ShEPhERD: Diffusing shape, electrostatics, and pharmacophores for bioisosteric drug design[C]//ICLR. 2025.

**Questions:**

* Why choose pharmacophore as a condition while consider ligand-based drug design? Could you give more justification?
* Why choose these four specific proteins? Could you compare on more protein targets, like CrossDocked2020[5]?
* Could you compare with more SBDD methods to make the claim 'higher docking score' more convincing?

[5] Francoeur P G, Masuda T, Sunseri J, et al. Three-dimensional convolutional neural networks and a cross-docked data set for structure-based drug design[J]. Journal of chemical information and modeling, 2020, 60(9): 4200-4215.

---

### Official Review · Reviewer_wnoh · 2025-10-27

**Soundness:** 3
**Presentation:** 2
**Contribution:** 2
**Rating:** 2
**Confidence:** 5

**Summary:**

A pharmacophore-conditioned SE(3)-equivariant diffusion model that feeds a pharmacophore graph into the generator and fuses it via hard inpainting (reinjecting constrained atoms each step) plus cross-attention to pharmacophore embeddings.
It adds a pharmacophore-weighted loss that explicitly enforces positions, types, and charges on pharmacophoric atoms.

**Strengths:**

Paper is overall  clear and easy to read, and follows the track of how molecule generative models were built. The figure explains the model architecture clearly and how generation is done. Pharmacophore based generation is a useful and practical approach in drug design.

**Weaknesses:**

1.the concept itself is not novel nor new, there has been many attempts following similar tactics. What makes your model special and different compared to the previous approaches( refer to the weakness 2)

2.there are many pharmacophore based generative models that has the codes to compare. an in-depth comparison between these models is necessary: what is strong and special with your model? what makes your model unique?

some models that i've found are: ShEPhERD (ICLR ’25), PhoreGen (Nat. Comput. Sci. ’25) DiffInt (ChemRxiv ’24) Hot2Mol (’24→’25) PP2Drug (arxiv “pharmacophore-constrained diffusion bridge”).

3. some metrics are low compared to recent SOTA models(validity is reaching 99% for some models, although they may not be pharmacophore conditioned)

4.  inpainting can lose key pharmacophoric features mentioned by authors. which leads to disconnected molecules (therefore low validity)

**Questions:**

1.as stated in the weakness, i think including the models above and comparing them would be necessary.
2.is freezing the conditions for pharmacophores necessary? what happens if we add some flexibility(noise) during generation. especially for the disconnected molecules. how about dropping a few pharmacophores to increase validity? maybe some pharmacophores are hard for the model to design.
3.in table 3 diffsbdd is a very old model. please replace to most recent models for better comparison.
4.can the authors release the code for reproduction?

---

### Official Review · Reviewer_m5BE · 2025-10-31

**Soundness:** 3
**Presentation:** 3
**Contribution:** 2
**Rating:** 2
**Confidence:** 3

**Summary:**

The authors present PharmaDiff, a pharmacophore-conditioned model for ligand generation.

**Strengths:**

All metrics are reported with error bars, which is highly appreciated.

I appreciate the well-documented instructions on github to replicate the results in the paper.

**Weaknesses:**

Comparisons to other methods are weak. Only other method compared to is DiffSBDD for structure-conditioned ligand generation. Additional comparisons to newer methods are needed, and on more than just 4 targets.

Methodological novelty is weak, as pharmacophore-conditioned generative ligand design has been shown (see “Pharmacophore-guided de novo drug design with diffusion bridge” by Wang and Rajapakse). Could the authors explain how their method differs from the one above?

**Questions:**

See Weaknesses.

“PharmaDiff aims to enable pharmacoophore-informed molecule generation without requiring the 3D structure of the protein target.” -- For how many proteins do we know the exact pharmacophores required, while also not having the 3D structure?

---

### Official Review · Reviewer_iLXt · 2025-10-31

**Soundness:** 3
**Presentation:** 3
**Contribution:** 2
**Rating:** 4
**Confidence:** 3

**Summary:**

The paper introduces PharmaDiff, an E(3)-equivariant pharmacophore-conditioned diffusion model for 3-D ligand generation without relying on protein structures. Instead of conditioning on atomic point clouds of binding pockets as in SBDD diffusion models, PharmaDiff directly takes a 3-D pharmacophore hypothesis—a spatial configuration of functional features such as hydrogen-bond donors, acceptors, hydrophobes, and aromatics—and generates molecules consistent with it.

contributions:

- Pharmacophore graph representation: encodes each pharmacophoric feature as an atom-level node with coordinates and physicochemical descriptors.

- Conditioned diffusion architecture: extends MiDi-style SE(3)-equivariant transformers with pharmacophore inpainting and cross-attention to enforce spatial constraints through the denoising trajectory.

- Pharmacophore-weighted loss for coordinate, atom-type, charge, and edge predictions to enhance feature fidelity.

- COM re-centering to preserve equivariance when re-injecting fixed pharmacophore atoms at every layer.

- Comprehensive experiments on GEOM-DRUGS and four protein targets showing superior pharmacophore match scores, docking performance, and drug-likeness versus state-of-the-art baselines (TransPharmer, REINVENT-4, PGMG, and DiffSBDD).

**Strengths:**

Clear motivation: bridges the gap between ligand-based and structure-based diffusion models by using pharmacophoric cues that are experimentally accessible yet geometry-aware.

Architectural novelty: the combination of inpainting, cross-attention, and equivariance-preserving alignment is well-engineered and principled.

Strong empirical evidence: PharmaDiff consistently improves pharmacophore-matching metrics (MS ≈ 0.90, PMR ≈ 0.70) and achieves better or comparable docking scores and synthetic accessibility to DiffSBDD.

Thorough comparisons and ablations: covers both ligand- and structure-based scenarios, demonstrating generality.

Reproducibility: detailed implementation, dataset splits, and open-source references are provided.

**Weaknesses:**

Ambiguity in conditioning signal: while pharmacophore features serve as conditioning inputs, it is unclear how stochastic or deterministic their placement is at inference—i.e., whether small coordinate perturbations affect generation stability.

Scalability limits: the method assumes a small number (3–7) of pharmacophoric points; it is uncertain how performance scales with larger or flexible hypotheses.

Ablation clarity: contributions of individual modules (cross-attention vs inpainting vs loss terms) could be quantified more precisely.

**Questions:**

Condition acquisition: In practical settings, how are pharmacophore hypotheses derived? Are they extracted from known ligands or predicted by external tools?

Generality: Can PharmaDiff handle partial or noisy pharmacophore inputs (e.g., missing coordinates or mislabeled features)?

Sampling efficiency: How sensitive is the model to the number of diffusion steps, and is inference time dominated by coordinate regression or discrete denoising?

Comparison fairness: DiffSBDD uses full protein structures; did the authors normalize conditions (e.g., number of fixed atoms) to ensure fair comparison?

Interpretability: Could the cross-attention weights between generated atoms and pharmacophore atoms provide interpretable mappings of functional alignment?

---

### Note · Authors · 2025-11-30

I have read and agree with the venue's withdrawal policy on behalf of myself and my co-authors.